# Machine learning nominal max oxygen consumption from wearable reflective pulse oximetry with density functional theory

**Saleem Abdul Fattah Ahmed Al Dajani** [1]  **Frédéric Laquai** [1]

## Abstract

Wearable sensors have revolutionized modern health towards precision medicine. Many measurements taken by wearables are not currently used for medical purposes, and their physical roots are not understood. Here we show the origin of inferring oxygen saturation from measuring estimated oxygen variation while preparing for a marathon via spectrophotometric pulsatile blood flow transcutaneous reflectance oximetry, building six years of data tracking the stages of running a marathon. Based on inputs from quantum mechanical simulations into classical wave and electromagnetic theory models, the imaginary part of the complex dielectric function of the active site of human hemoglobin is altered upon oxygen binding to the heme group, changing the extinction coefficient and selectively absorbing wavelengths of light in a way that enables its detection from the ratio of red to infrared absorbance. A fundamental nominal max oxygen consumption is inferred from quantum mechanical absorbance spectra with and without oxygen binding by training a machine learning model on 1.5 years of daily wearable reflective pulse oximetry data with an $R^2$ of 0.84. Reflectance oximetry is strongly correlated ($R^2$ of 0.96) to standard pulse oximetry, such as through earlobe or fingertip, enabling accurate, rapid, regular, and automated monitoring of vital signs with wearable sensors on smart watches and fitness trackers, and supplying artificial intelligence inferences of function-symptom relationships for precision medicine.

[1]Applied Physics (AP) Program, Extreme Computing Research Center (ECRC), and KAUST Solar Center (KSC), Physical Science and Engineering (PSE) and Computer, Electrical, Mathematical Sciences and Engineering (CEMSE) Divisions, King Abdullah University of Science and Technology (KAUST), Thuwal, Makkah Province, Kingdom of Saudi Arabia (KSA) 23955-6900. Correspondence to: Saleem Abdul Fattah Ahmed Al Dajani <saleem.abdulfattah.aldajani@gmail.com>.

*Accepted at the 1st Machine Learning for Life and Material Sciences Workshop at ICML 2024.* Copyright 2024 by the author(s).

## 1. Introduction & Background

Recent breakthroughs in artificial intelligence have paved the way for precision medicine in unprecedented ways (Johnson, Kevin B et al., 2021; Mesko, 2017; Subramanian, Murugan et al., 2020). The democratization and accessibility of machine learning algorithms in standard computing libraries have enabled the usage of advanced analytics to analyze large datasets that are challenging to analyze with conventional methods like regression. In this study, random forest regression (Breiman, 2001; Smith et al., 2013) is implemented to machine learn nominal max oxygen consumption rate, $\dot{V}O_{2,max}$ from commercial wearable sensor data based on density functional theory simulations of hemoglobin in oxygenated and deoxygenated states.

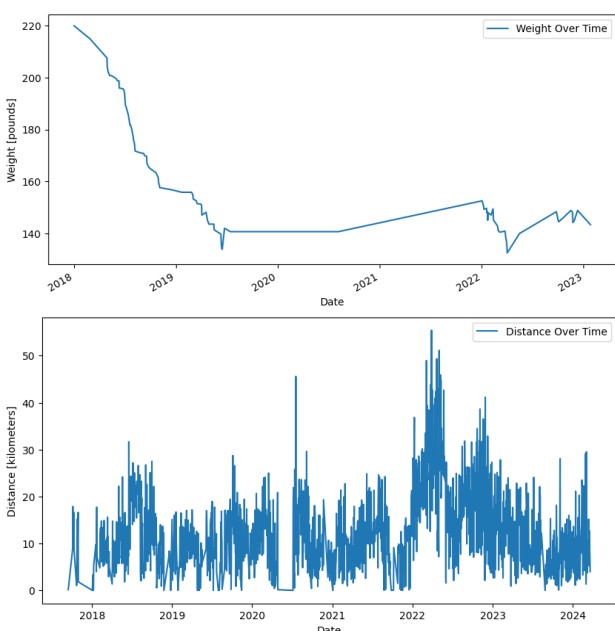

*Figure 1.* Time-series data for (top) weight loss and (bottom) distance for 5-6 years throughout marathon running in March 2022 with preparation prior and consistent running after.

Pulse oximetry is a standard physiological measure of oxygen saturation, and is typically conducted by fingertip or earlobe tests (Mendelson et al., 1983; Hoffman, 2017). More

recently, reflective pulse oximetry systems (Li, 2010) have been developed and miniaturized to be conducted using LEDs on wearable fitness trackers, such as Fitbit, Apple Watch, WHOOP, Oura rings, and other platforms (Snyder Lab at Stanford University, 2024; White et al., 2023). These systems measure oxygen variation as the ratio of red to infrared ratios reflected off skin. Using density functional theory, under the independent particle assumption, Fermi's golden rule is employed to calculate absorption coefficients from which the absorbance ratio could be calculated from first principle simulations (Iurii Timrov, Oscar Baseggip). Life and material science could be integrated in this way from theory to industry applications with hemoglobin as an example, where the methods could be extended to other biological systems and signals from similar or other material properties that could be calculated from the solution to the Schrödinger many-body wave equation, posed as an Euler-Lagrange formulation of the electron density ground state energy eigenvalue problem (Schwingenschlögl, 2004):

$$\hat{H}\psi(\mathbf{r}) = -\frac{\hbar^2}{2m}\nabla^2\psi(\mathbf{r}) + V(\mathbf{r})\psi(\mathbf{r}) = E\psi(\mathbf{r}) \quad (1)$$

where $\psi(\mathbf{r})$ is the wave function of the particle, dependent on the position vector $\mathbf{r} = (x, y, z)$, $\nabla^2$ is the Laplacian operator, $\nabla^2 = \frac{\partial^2}{\partial x^2} + \frac{\partial^2}{\partial y^2} + \frac{\partial^2}{\partial z^2}$, $\hbar$ is the reduced Planck's constant, $E$ is the energy eigenvalue associated with the state $\psi(\mathbf{r})$. These theorems establish the basis for using electron density as a principal variable in determining the ground state properties of a many-electron system (Schwingenschlögl, 2004):

**Theorem 1** (Ground state electron density uniqueness, $\rho_0$). *Let $\Psi_0$ be the ground state wavefunction of a many-electron system with external potential $V_{ext}$ and Hamiltonian $\hat{H}$. The ground state electron density $\rho_0(\mathbf{r})$, defined by $\rho_0(\mathbf{r}) = N \int |\Psi_0(\mathbf{r}, \mathbf{r}_2, \ldots, \mathbf{r}_N)|^2 \, d\mathbf{r}_2 \ldots d\mathbf{r}_N$, uniquely determines $V_{ext}(\mathbf{r})$ up to an additive constant.*

**Theorem 2** (Energy functional variational principle, $E[\rho]$). *The ground state energy $E_0$ of a many-electron system is a unique functional of the electron density $\rho$, denoted $E[\rho]$. Among all possible electron densities, the true ground state density $\rho_0$ minimizes $E[\rho]$ (Schwingenschlögl, 2004):*

$$E[\rho_0] = \min_\rho E[\rho]. \quad (2)$$

The Kohn-Sham approach to density functional theory (DFT) involves solving a set of non-interacting particles that mimic the behavior of real, interacting particles. The total energy functional in Kohn-Sham DFT is given by the application of the Kohn-Sham Hamiltonian to the ground state electron density (Schwingenschlögl, 2004):

$$E[\rho_0] = T_s[\rho_0] + \int V_{ext}(\mathbf{r})\rho_0(\mathbf{r}) \, d\mathbf{r} \quad (3)$$

$$+ \frac{1}{2} \int \int \frac{\rho_0(\mathbf{r})\rho_0(\mathbf{r}')}{|\mathbf{r} - \mathbf{r}'|} \, d\mathbf{r} \, d\mathbf{r}' + E_{xc}[\rho]$$

where $T_s[\rho]$ is the kinetic energy of the non-interacting reference system, $V_{ext}(\mathbf{r})$ is the external potential energy, the third term represents the classical Coulomb interaction of the electron density, $E_{xc}[\rho]$ is the exchange-correlation energy functional. To minimize the energy functional with respect to the density $\rho$, under the constraint of normalized wavefunctions, we introduce the Kohn-Sham orbitals $\psi_i(\mathbf{r})$ where $\rho(\mathbf{r}) = \sum_{i=1}^{N} |\psi_i(\mathbf{r})|^2$. The variation of the energy functional with respect to $\psi_i^*(\mathbf{r})$ leads to the Kohn-Sham equations by Euler-Lagrange formulation (Schwingenschlögl, 2004):

$$\left(-\frac{\hbar^2}{2m}\nabla^2 + V_{eff}(\mathbf{r})\right)\psi_i(\mathbf{r}) = \epsilon_i\psi_i(\mathbf{r}) \quad (4)$$

where $\epsilon_i$ is the Lagrange parameter approximation of the energy eigenvalue associated with the state $\psi_i$, and $V_{eff}(\mathbf{r})$ is the effective potential given by (Schwingenschlögl, 2004):

$$V_{eff}(\mathbf{r}) = V_{ext}(\mathbf{r}) + \int \frac{\rho(\mathbf{r}')}{|\mathbf{r} - \mathbf{r}'|} \, d\mathbf{r}' + V_{xc}(\mathbf{r}) \quad (5)$$

and $V_{xc}(\mathbf{r})$ is the functional derivative of the exchange-correlation energy $E_{xc}[\rho]$ with respect to the density, $\frac{\delta E_{xc}}{\delta \rho}$. The ground state electron density $\rho_0(\mathbf{r})$ is calculated by summing the square of the absolute value of each occupied Kohn-Sham orbital, where $N$ is the number of electrons (or occupied states) (Schwingenschlögl, 2004):

$$\rho_0(\mathbf{r}) = \sum_{i=1}^{N} |\psi_i(\mathbf{r})|^2 \quad (6)$$

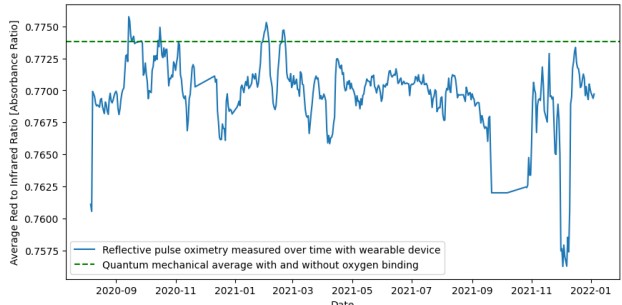

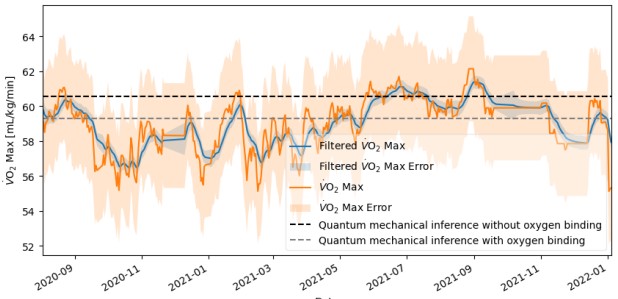

*Figure 2.* Time-series data for (top) reflective pulse oximetry and (bottom) max oxygen consumption rate for 1.5 years while preparing for marathon in March 2022 for nominal conditions prior.

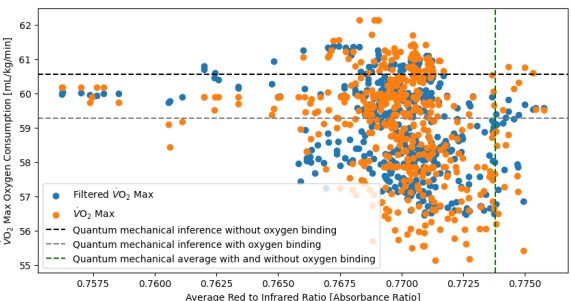

*Figure 3.* Scatter of raw data of absorbance ratio versus max oxygen consumption rate for 1.5 years while preparing for marathon in March 2022 for nominal conditions prior.

## 2. Methods & Datasets

Wearable sensor data was collected using Fitbit Alta HR and Fitbit Charge 4 over the course of marathon running and preparation from 2018 up to 2024, including weight, distances, absorbance ratio, max oxygen consumption rate, and other measurements. Summary of the entire dataset is shown in Fig. 1, and the subest analyzed in this study is shown separately in Fig. 2 and in a combined scatter plot in Fig. 3. These datasets are read with standard computing libraries for parsing comma-separated text files.

Density functional theory was performed in Quantum ESPRESSO for hemoglobin in oxygen and deoxygenated state with standard structural relaxation using BFGS-embedded self-consistent field calculation loops, represented mathematically as follows (Hung, Nguyen Tuan, Nugraha, Ahmad R. T., and Saito, Riichiro, 2022):

$$\underset{x \in R^n}{\arg\min} f(x) | \nabla f(x^*) = 0 | H_0^{-1} \approx \nabla^2 f(x_0) \quad (7)$$

$$x_{k+1} = x_k - H_k^{-1} \nabla f(x_k) \quad (8)$$

$$s_k = x_{k+1} - x_k \quad (9)$$

$$y_k = \nabla f(x_{k+1}) - \nabla f(x_k) \quad (10)$$

$$H_{k+1} = H_k - \frac{H_k s_k s_k^T H_k}{s_k^T H_k s_k} + \frac{y_k y_k^T}{y_k^T s_k} \quad (11)$$

Outputs from structural relaxation were then used as inputs to the epsilon.x framework for the dielectric tensor, which is then used to calculate extinction coefficient, refractive index, absorption coefficient, and other optical properties (Iurii Timrov, Oscar Baseggip). The complex dielectric function can be expressed as (Iurii Timrov, Oscar Baseggip):

$$\epsilon(\omega) = \epsilon_1(\omega) + i\epsilon_2(\omega) \quad (12)$$

where $\epsilon_1(\omega)$ is the real part of the dielectric function, and $\epsilon_2(\omega)$ is the imaginary part which is related to the absorption of electromagnetic waves by the material. The imaginary

part of the dielectric function, $\epsilon_2(\omega)$, can be computed from the electronic structure using the following relation based on the Fermi's golden rule:

$$\epsilon_2(\omega) = \frac{8\pi^2 e^2}{\Omega \omega^2} \sum_{v,c,\mathbf{k}} |\langle \psi_{c\mathbf{k}} | \hat{\mathbf{p}} | \psi_{v\mathbf{k}} \rangle|^2 \delta(E_{c\mathbf{k}} - E_{v\mathbf{k}} - \hbar\omega) \quad (13)$$

where $e$ is the elementary charge, $\Omega$ is the volume of the unit cell, $\psi_{c\mathbf{k}}$ and $\psi_{v\mathbf{k}}$ are the conduction and valence band wavefunctions at wavevector $\mathbf{k}$, $E_{c\mathbf{k}}$ and $E_{v\mathbf{k}}$ are the energies of these states, $\hat{\mathbf{p}}$ is the momentum operator, and $\delta$ is the Dirac delta function ensuring energy conservation. The real part, $\epsilon_1(\omega)$, can be obtained from $\epsilon_2(\omega)$ using the Kramers-Kronig relations, where $\mathcal{P}$ denotes the principal value of the integral:

$$\epsilon_1(\omega) = 1 + \frac{2}{\pi}\mathcal{P} \int_0^\infty \frac{\xi \epsilon_2(\xi)}{\xi^2 - \omega^2} d\xi \quad (14)$$

Analysis of DFT outputs is done using standard spreadsheet analysis tools. The absorption coefficient $\alpha(\omega)$ is calculated from the imaginary part of the dielectric tensor using the formula (Dresselhaus, 2001; Fox, 2010):

$$\alpha(\omega) = \frac{2\omega\kappa(\omega)}{c} = \frac{4\pi\kappa(\omega)}{c} \quad (15)$$

where $\kappa(\omega)$ is the extinction coefficient. The refractive index $n(\omega)$ is related to the complex dielectric function by:

$$n^2(\omega) = \epsilon(\omega) = \epsilon_1(\omega) + i\epsilon_2(\omega) \quad (16)$$

from which $\kappa(\omega)$ is derived as:

$$\kappa(\omega) = \sqrt{\frac{\sqrt{\epsilon_1(\omega)^2 + \epsilon_2(\omega)^2} - \epsilon_1(\omega)}{2}} \quad (17)$$

Substituting $\kappa(\omega)$ into the absorption coefficient formula provides the measure of how much light is absorbed by the material as a function of frequency, which is converted to wavelength by $\lambda = \frac{hc}{\omega}$ for $\alpha(\lambda)$ shown in Fig. 4.

---

**Algorithm 1** Random Forest Regressor (Breiman, 2001)

**Input:** Training data $X$ (features) in Fig. 2 and $Y$ (targets) in Fig. 2, number of trees $N$, tree depth $D$
**Output:** Random Forest model $M$
Initialize forest $F$ with $N$ empty trees
**for** $n = 1$ **to** $N$ **do**
    $X_n, Y_n \leftarrow$ Bootstrap sample from $X, Y$ in Fig. 3
    Initialize tree $T_n$ with depth $D$
    Build $T_n$ using $X_n, Y_n$ in Fig. 2
    Add $T_n$ to forest $F$ according to Eqn. 18
**end for**
$M \leftarrow$ Aggregate predictions from all trees in $F$
**return** $M$

---

Machine learning is performed with the sklearn ensemble learning libraries for random forest regression, which is

described in Alg. 1 (Breiman, 2001) and is mathematically represented as:

$$\hat{y}(x) = \frac{1}{n} \sum_{i=1}^{n} T_i(x) \quad (18)$$

where $\hat{y}(x)$ is the predicted value for the input $x$, $n$ is the number of trees in the forest, $T_i(x)$ is the output prediction of the leaf node reached by traversing tree $i$ using decisions based on $x$, such that:

$$T_i(x) = \begin{cases} v_{i1} & \text{if } x \text{ satisfies condition } C_{i1} \\ v_{i2} & \text{if } x \text{ satisfies condition } C_{i2} \\ \vdots \\ v_{ik} & \text{if } x \text{ satisfies condition } C_{ik} \end{cases} \quad (19)$$

where $v_{ij}$ represents the output at each leaf node in the $i$-th tree, $C_{ij}$ represents the conditions based on the input features that guide the traversal to each leaf node, $k$ is the number of leaf nodes in tree $i$.

## 3. Results

Weight loss throughout preparation and marathon running between 2018 and 2024 are shown in Fig. 1. Weight loss begins linearly and then stabilized within the ideal body mass index range based on weight and height. Distance begins at an average of ten kilometers per day, and then ten kilometers twice a day, followed by ten kilometers three times a day. Closer to the marathon date, these ten-kilometer runs were combined into 20-30km runs once or twice a day. Absorbance ratio and max oxygen consumption measurements in the preparation phase between September 2020 and January 2022 are shown in Fig. 2. The quantum absorbance ratio and quantum inference are in agreement with the physiological measurements.

*Table 1.* Summary of density functional theory results using the independent particle assumption-based absorption coefficient from transition state probabilities by Fermi's golden rule based on the ground state electron density dielectric tensor (Iurii Timrov, Oscar Baseggip), $\epsilon_\rho$, where $\alpha_x$ is the $x$−nanometer absorption coefficient and $\frac{A_X}{A_Y}$ is the X-Y-nanometer absorbance ratio.

| Calculated property | Hemoglobin (Hb) | Oxyhemoglobin (HbO2) |
|---|---|---|
| $\alpha_{660nm}\left[\frac{1}{cm}\right]$ | $6.3\times10^3$ | $4.8\times10^3$ |
| $\alpha_{940nm}\left[\frac{1}{cm}\right]$ | $8.8\times10^3$ | $5.7\times10^3$ |
| $\frac{\alpha_{660nm}}{\alpha_{940nm}}$ $\left(\frac{A_{660nm}}{A_{940nm}}\right)$ | 0.71 | 0.84 |

DFT absorption coefficients are shown in the upper panel of Fig. 4 for hemoglobin in the deoxygenated and oxygenated states based on the structures without oxygen binding in the center-lower left panels and with oxygen binding in the center-lower right panels. The absorption coefficients and absorbance ratios are summarized in Tab. 1.

Prediction-powered inference bounds (Angelopoulos, Anastasios N et al., 2023) were calculated by training the same models on the measurement error with the same DFT values of absorbance ratio as inputs. These quantum absorbance ratios are in agreement with the raw data in Fig. 2 and in the scatter plot in Fig. 3 and shown as dashed lines.

Ensemble machine learning with a random forest regressor, summarized in Alg. 1, is compared to standard linear regression in Fig. 5. The linear regression shows a poor $R^2$ value of 0.11-0.19 while the machine learning model shows a superior $R^2$ value of 0.83-0.84. Inferences based on DFT inputs of the absorbance ratios in Tab. 1 are summarized in Tab. 2. These quantum inferences match measurements shown in Fig. 2 and scatter plot shown in Fig. 3.

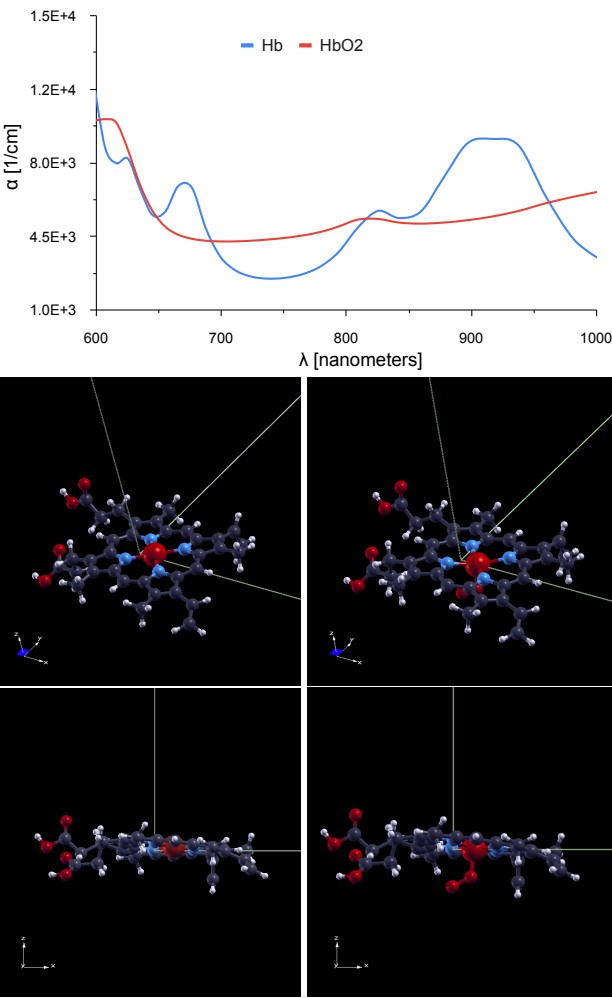

*Figure 4.* Quantum mechanical (top) absorption coefficient for the active site of human hemoglobin before (center-bottom left) and after binding oxygen (center-bottom right) at planar and side-views of the heme group.

*Table 2.* Summary of machine learning results. Asterisks refers to $R^2$ for standard and reflective pulse oximetry correlation (Mendelson et al., 1983). Prediction-powered inference errors (Angelopoulos, Anastasios N et al., 2023) are calculated by the same models trained on the measurement errors shown in Fig. 2.

| Density functional theory prediction-powered inference | Hb, A660/A940=0.71 | HbO2, A660/A940=0.84 | $R^2$ value |
|---|---|---|---|
| SpO$_2$/SaO$_2$ % | 96.44 | 93.15 | 0.96* |
| Linear $\dot{V}O_2$ max | 69.07 ± 8.63 | 47.35 ± 12.51 | 0.11 |
| Random forest $\dot{V}O_2$ max | 60.43 ± 3.00 | 58.97 ± 3.00 | 0.83 |
| Linear filtered $\dot{V}O_2$ max | 69.77 ± 8.63 | 46.65 ± 10.50 | 0.19 |
| Random forest filtered $\dot{V}O_2$ max | 60.71 ± 0.82 | 59.59 ± 2.28 | 0.84 |

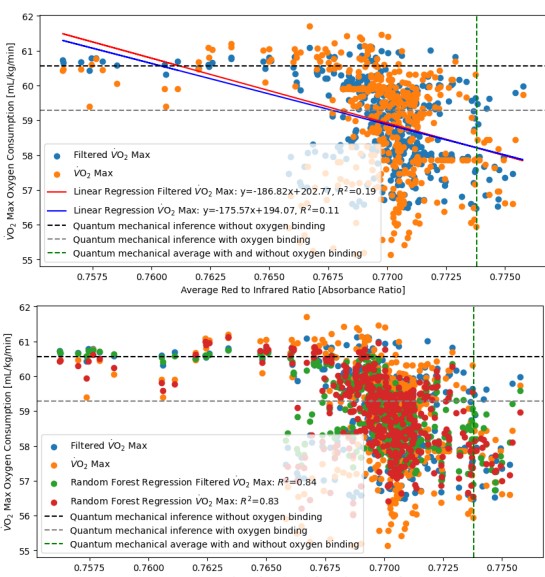

*Figure 5.* Machine learning max oxygen consumption from wearable reflective pulse oximetry with density functional theory, comparing (top) linear regression with (bottom) random forest regression to train model on wearable data for 1.5 years.

## 4. Discussion

As a result of weight loss and increased mileage up to 25,000 kilometers total throughout the duration of the study, the absorbance ratio evidently decreases and the max oxygen consumption increases by about 2-4 mL/kg/min as shown in Fig. 1 and Fig. 2. At steady state, this is attributed to the decrease in oxygenated hemoglobin and increase in deoxygenated hemoglobin due to increased cardiovascular fitness, where lower $\frac{A_{660}}{A_{940}}$ absorbance ratio implies lower A660 in oxygenated hemoglobin, and/or higher A940 in deoxygenated hemoglobin (Fitbit Help Center, 2024) as shown in Fig. 4, therefore, reflecting more red light than infrared and absorbing more infrared, making blood red.

The DFT absorbance ratio appears to be an upper limit on the measured absorbance ratio, shown as the maximum absorbance ratio in Fig. 2. This is because the model only considers the active unit as one heme group either bonded or not bonded to oxygen, when in reality, hemoglobin has four active sites with increasing cooperativity of oxygen binding.

The inferred quantum max oxygen consumption rate, $\dot{V}O_{2,max}^{quantum}$, is in agreement with measurements shown in Fig. 2. The trend is clearly not linear, as shown in Fig. 5, but certainly within the bounds calculated and inferred from DFT as shown in Fig. 3. To achieve greater than 90% accuracy, the model must be enhanced to incorporate the noise in the data, which is likely the reason why the $R^2$ drops to 0.83-0.84. Upon inspection of the lower panel of Fig. 2, the inference is in agreement with the measurements.

The first principles absorbance ratios shown in Fig. 4 are in general agreement with experimental values (Mendelson et al., 1983; Hoffman, 2017), however, some features are missing, likely due to the simplified single active unit model of hemoglobin. Only one oxygen binding orientation is considered, when both sides are possibilities depending on steric hindrance. Future studies could include all four active units upon scaling DFT algorithms to include higher atom counts, since the single active unit is already bordering the computational limitation of hundreds of atoms.

## 5. Conclusion

This work shows that first principles calculations powered by wearable sensor data could enable AI-powered precision medicine. Hemoglobin is demonstrated as an example in this study and its compatibility with first principles calculation-based inferences through wearable sensor measurement models trained on physiological quantities.

Future work could extend the approach to other biological systems and signals, such as glucose and insulin measurements. Similar simplified DFT models for active sites of biomolecules could be constructed, and wearable sensor physiological measurements could be calculated and used for inferences of nominal quantities of the physiological function of interest.

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
