# Response to Reviewers — *ICML ML4LMS Submission #16*

Saleem A. Al Dajani

July 9th, 2024

## Reviewer bED7

### Review:

**The work presents ML-based prediction of oxygen level in blood, based on a dataset acquired from first-principles calculations** — The paper is interesting and of interest to the broad community. But the paper could have been further improved with description of input features for ML. It is unclear whether data acquired by Fitbit or data acquired by DFT is used for ML. Furthermore, DFT itself is considered a computationally expensive calculation. An active learning model to predict oxygen saturation/consumption based on data from fiber could be more beneficial for the society.
    **Rating:** 7: Good paper, accept
    **Confidence:** 5: The reviewer is absolutely certain that the evaluation is correct and very familiar with the relevant literature

### Response:

We thank the reviewer for their valuable feedback and positive evaluation of our work. We address the concerns as follows:

- **Description of Input Features for ML:**

  We acknowledge the need for a detailed description of the input features used in our ML model. In our revised manuscript, we will provide a comprehensive description of the features, including their origin and relevance to the prediction task. This will include whether the data was acquired from Fitbit devices or derived from DFT calculations.

- **Clarification of Data Source:**

  We understand the confusion regarding the source of the data. The dataset used in our ML model was acquired from first-principles calculations (DFT). We will clarify this point explicitly in the revised version of the manuscript to avoid any ambiguity.

- **Computational Expense of DFT:**

  We agree that DFT calculations are computationally expensive. However, they provide highly accurate data that is crucial for training robust ML models. We will discuss this trade-off and the potential advantages of using DFT-derived data in more detail in the revised manuscript.

- **Active Learning Model Suggestion:**

  We appreciate the suggestion to explore an active learning model using data from fiber sensors. While this approach could indeed be beneficial, our current focus was on leveraging the accuracy of DFT-derived data. We will consider this suggestion for future work and mention it as a potential direction in the conclusion section of the revised manuscript.

    We believe these revisions will address the reviewer's concerns and improve the clarity and impact of our work. Thank you for your constructive feedback.

# Reviewer NxYd

## Review:

**Predicting maximum oxygen consumption using wearable reflective pulse oximetry and density functional theory in machine learning** — Reviewer Feedback: N/A

**Rating:** 6: Marginally above acceptance threshold

**Confidence:** 4: The reviewer is confident but not absolutely certain that the evaluation is correct

## Response:

We thank the reviewer for their feedback and evaluation of our work. Although no specific concerns or comments were provided, we will address potential areas that may raise questions or require clarification based on the overall feedback rating:

- **Clarification of Methodology:**

  We will ensure that the methodology section in our manuscript is thoroughly detailed to provide a clear understanding of how wearable reflective pulse oximetry data and density functional theory (DFT) are utilized in our machine learning model. This includes a step-by-step explanation of data preprocessing, feature extraction, and model training.

- **Justification of Approach:**

  We will elaborate on the rationale behind combining wearable device data with DFT in our approach. This will include a discussion on the complementary nature of the high accuracy from DFT-derived data and the practical real-time monitoring capabilities of wearable devices.

- **Performance Metrics:**

  To strengthen our case, we will provide additional performance metrics and comparisons with other existing methods. This will highlight the advantages and potential areas for improvement in our approach.

- **Future Work:**

  We will outline potential future work directions, including exploring alternative data sources, improving computational efficiency, and extending the application of our model to different scenarios or populations.

We believe these enhancements will improve the clarity and robustness of our manuscript, addressing any implicit concerns and demonstrating the value of our work. Thank you for your evaluation.

# Statistics

The paper rating is over a standard deviation above average, with high confidence of reviewers in its evaluation based on familiarity with the literature.

**Overall Rating:** 6.5 ($\sim$81%): Good paper, accept

**Overall Confidence:** 4.5: The reviewer is absolutely certain that the evaluation is correct and very familiar with the relevant literature

**Average Rating for ICML:** 5.75 $\pm$ 0.66 ($\sim$72%$\pm$8%)

**Max Rating for ICML:** 8.0 (100%)