# OpenReview forum: "Machine learning nominal max oxygen consumption from wearable reflective pulse oximetry with density functional theory"
_ICML.cc/2024/Workshop/ML4LMS — ML4LMS Poster_

### Official Review · Reviewer_NxYd · 2024-06-11
**Predicting maximum oxygen consumption using wearable reflective pulse oximetry and density functional theory in machine learning.**

**Rating:** 6
**Confidence:** 4

**Review:**

N/A

---

### Official Review · Reviewer_bED7 · 2024-06-12
**The work presents ML-based prediction of oxygen level in blood, based on dataset acquired from first principles calculations.**

**Rating:** 7
**Confidence:** 5

**Review:**

The paper is interesting and of interest to the broad community. But the paper could have been further improved with description of input features for ML. It is unclear whether data acquired by Fitbit or data acquired by DFT is used for ML. Furthermore, DFT itself is considered computationally expensive calculation. An active learning model to predict oxygen saturation /consumption based on data from fiber could be more beneficial for the society.